# Investigation on the Bending Behavior of Tea Stalks Based on Non-Prismatic Beam with Virtual Internodes

Wenchao Wu, Yongguang Hu * and Zehui Jiang

Key Laboratory of Modern Agricultural Equipment and Technology, Ministry of Education Jiangsu Province, Jiangsu University, Zhenjiang 212013, China; 2111716005@stmail.ujs.edu.cn (W.W.); 2222016029@stmail.ujs.edu.cn (Z.J.)
* Correspondence: deerhu@ujs.edu.cn; Tel.: +86-138-1515-1176

**Abstract:** The study aims to fully explicate the bending behavior of tea stalks under the condition of large deflection, which is crucial to improve the working performance of mechanized harvesting equipment. The mechanical model of the stalk was assumed to be a non-prismatic beam with virtual internodes that could differ from actual internodes. With the model, the stalk can be freely divided into multiple virtual internodes, whose flexural rigidities can be determined by solving an optimization problem, and deflection curves can be predicted after determining the positions of virtual nodes under given loads. Moreover, a novel method was proposed to obtain the deflection curve of the stalk based on the techniques of binocular vision and non-uniform rational B-spline (NURBS) curve fitting. The results show that R-squared values of fitted 2nd-degree NURBS curves of bending shape of tea stalks range from 0.9576 to 0.9964, with an average of 0.9797. The results indicate that flexural rigidity decreases from the bottom to the top of the tea stalk, and the deflection curve could be predicted more precisely with the model of piecewise flexural rigidities than that of average flexural rigidity. The study could be applied to the optimization design of the cutter and adaptive adjustment techniques of operational parameters for tea picking machines.

**Keywords:** flexural rigidity; deflection; non-uniform rational B-spline; optimization

## 1. Introduction

Labor shortage and low mechanization have become huge challenges for tea harvest in China [1–3], it is urgent to develop efficient and reliable mechanized tea picking technology. Commonly used tea picking machines [4] harvest tea shoots with the cutter composed of double reciprocating multi-teeth blades like combine-harvesters, but the machine usually results in poor quality of harvested tea shoots due to broken leaves and bad cut surface on the stem. Therefore, it is necessary to analyze interaction between the tea stalk and the cutter for understanding what determines the working performance of the machine. During the harvesting process, the tea stalk is first bended and then clamped and cut off by the teeth of the blades. The process could be separated into two phases: (1) bending and (2) clamping and cutting. Essentially, the bending shape of the stalk and the cut position on the stalk are vital factors that together determine whether the harvested tea shoot can satisfy the processing needs. Hence, it is significant to study the bending behavior of tea stalks to design or optimize the cutter and determine operational parameters such as cutting height, forward speed, and cutting frequency for the machine.

Many studies have been conducted to study relationships between the bending properties and physical properties of plant stalks. Xue and Cao [5] studied the influences of internode position and tea variety on the elastic modulus and breaking deflection of tea stalks. Du et al. [6] analyzed the grey correlations and established regression models between physical properties, including stem segment number, stem diameter, moisture content, cellulose content, lignin content and so on, and the elastic modulus of tea stalks.

İnce et al. [7] analyzed the bending properties of sunflower stalk residue under four moisture contents at the lower, middle, and upper regions, respectively. They found that the elastic modulus of the stalk residue decreased with an increase in moisture and stalk diameter, whose effects decreased at lower moisture contents. Ahmad et al. [8] and Tavakoli et al. [9] found that the elastic modulus of wheat stalks decreased from the first internode to the third internode down from the ear, and the elastic modulus was negatively correlated with the moisture content. Moreover, similar results were also reported for alfalfa stalk [10], safflower stalk [11,12], rice stalk [13], and barley stalk [14].

In most studies, plant stalks are cut into multiple segments at first; then, the moment of inertia of the stalk's segment is calculated by regarding its cross section as a solid or hollow circle [15] or ellipse [16,17]; next, either three-point bending test or cantilever bending test are performed on the stalk's segment by universal test machine, texture analyzer, or self-established experimental system; finally, the average value of elastic modulus of the stalk's segment is calculated by approximate equations of the deflection curves of simply supported beam or cantilever beam subjected to a concentrated load. However, it is difficult to make plant stalks, such as tea stalks, rice stalks and wheat stalks, into ideal test specimens with a standard geometric size, because the cross section is irregular and varied along the axial line of plant stalks. Moreover, the approximate equations, referring to the book edited by Gere and Timoshenko [18], can only be used when the deflection curves of plant stalks have very small rotation angles, very small deflections, and very small curvatures, which means the rotation angle (in radians) is approximately equal to the slope for the deflection curve. Therefore, big errors may occur if separately calculating the moment of inertia and elastic modulus of plant stalks.

Inoue et al. [19] derived the differential equation of deflection curve of crop stalks based on cantilever beam with constant flexural rigidity, subject to concentrated load and large deflection, and a method was proposed to get the nodes' coordinates by line-shift camera and image processing. On this basis, Hirai et al. [20] proposed a method for calculating the flexural rigidity of crop stalks by the mechanical model based on the actual structure of crop stalks, with no need for cutting crop stalks into small segments. However, multiple markers must be bonded to a crop stalk for obtaining the coordinates of actual nodes, and the flexural rigidities of all internodes were solved by the trial-and-error method one by one. Moreover, calculation for flexural rigidity of an internode only considers the deflection at the corresponding node; therefore, the calculated flexural rigidity could not accurately represent the average flexural rigidity of the whole internode, and the longer the internode is, the greater the error would be. The mechanical model based on the actual structure of crop stalks [20] was further applied to analyze the dynamic responses of the reaction force and bending posture of rice stalk and wheat stalk [21–24].

The bending behavior of plant stalks is the combined effects of external load and self-load caused by the weight of the stalk itself, but the self-weight of plant stalks is often neglected. Fukushima and Sato [25] established a mechanical model for cabbage hypocotyl to clarify the deformation due to weight of the part above the cotyledonary node. The hypocotyl was regarded as a prismatic beam with constant flexural rigidity, and the results indicate that the flexural rigidity of the hypocotyl increases exponentially proportional to the number of days elapsed during the seedling stage. Stubbs et al. [26] investigated the effect of plant weight on estimations of stalk lodging resistance, and they found that a significant error would occur when ignoring the effect of self-load on wheat and rice stalks, which have relatively larger ears, whereas no significant errors occurred for large and stiff plants such as maize, bamboo, and sorghum. However, plant stalks were simplified as a prismatic beam with constant flexural rigidity, and approximate equations of deflection curves of beams under the condition of small deflection were used in the study.

Few literatures have been published on the bending behavior of tea stalks. There has been no suitable and convenient method to obtain the deflection curve of plant stalks until now. Plant stalks are usually simplified as either a prismatic beam with constant flexural rigidity or a non-prismatic beam, in which each actual internode was regarded as a

prismatic beam with constant flexural rigidity. Although the latter could better characterize the bending behavior of plant stalks in mechanics than the former, it is also limited by the actual internodes of plant stalks. In fact, flexural rigidity varies along the axial line of plant stalks even in the same internode, because of different maturity levels, varied cross-sectional shapes, and the complicated biological tissues of plant stalks.

In this study, we focused on investigating the bending behavior of individual tea stalk to provide a fundamental basis for the optimization design of the cutter and adaptive adjustment techniques of the operational parameters for tea picking machines. A novel method was proposed to obtain the deflection curve of tea stalk based on the binocular vision technique and non-uniform rational B-spline (NURBS) curve fitting technique. Moreover, a mechanical model was developed for tea stalk, and the concepts of virtual node and virtual internode were introduced in the model. With the model, two methods were developed to solve the piecewise flexural rigidities of virtual internodes and predict the deflection curve of tea stalk under given loads, respectively.

## 2. Materials and Methods

### 2.1. Materials and Experimental System

Tea stalk samples were collected in Maichun tea plantation at Danyang, Jiangsu province, China ($32°02'35''$ N, $119°67'80''$ E). Two varieties of tea plants, *Zhongcha* 108 and *Maolv*, were selected for the study, and a 20-m-long sampling zone of each tea species was pruned on 15 July 2021. After 40 days of growth, 10 straight stalks were collected from each species on September 5. All leaves were removed during the experiment.

An experimental system was designed for the bending test, as shown in Figure 1. The system consists of horizontal testbed, clamp, lift platform, horizontal hand-operated stand, ZP-10 digital force meter (Fuma Electric Equipment Co., Ltd., Dongguan, Guangdong, China), personal computer, ZED 2 stereo camera (Stereolabs Inc., San Francisco, CA, USA), and a tripod. The digital force sensor was mounted on the horizontal hand-operated stand, which was placed on the lift platform and could drive the digital force sensor forward or backward by rotating its shaft. The bottom end of the tea stalk can be fixed with the clamp, and the body of the tea stalk can be deflected by the digital force meter. The ZED 2 camera was mounted on the tripod, and it was controlled by the personal computer. The camera has four available resolutions, and the resolution $4416 \times 1242$ was chosen for this study. All components of experimental system, except for the camera and tripod, were placed on the horizontal testbed. A preliminary test should be conducted to adjust the tripod head to ensure that the camera can capture the shape of tea stalks during experiment, after that, neither the camera nor the tripod can be moved.

### 2.2. Determination of Deflection Curve of Tea Stalks

2.2.1. Data Acquisition and Preprocessing

Each tea stalk was bended at six heights with random deflections, including a situation in which the load acts on the third internode, which is the ideal picking position for general tea materials. Tea stalks were fixed with the clamp, and the stalk length was measured and recorded. The initial shape or bending shape of the tea stalk could be captured by the stereo camera, and a point cloud, which describes the shape of the tea stalk, could be obtained by the following steps. Firstly, background subtraction, a technique for foreground detection in the field of computer vision, was applied to extract tea stalk in the left image and right image captured by the stereo camera. Then, a point cloud can be generated by computing the stereo disparity between the left image and right image. Finally, noise reduction was done, and the point cloud of tea stalk could be obtained.

Moreover, some operations were applied on the point clouds of tea stalks. At first, point clouds belonging to the $i$th tea stalk are put together to form a point set $G_i^0 = \left\{ G_{i_0}^0, \ G_{i_1}^0, \ \cdots, \ G_{i_j}^0, \ \cdots \right\}$, where $G_{i_0}^0$ and $G_{i_j}^0$ are the point clouds for the initial shape and the $j$th bending shape of the $i$th tea stalk, respectively. Then, principal component analysis is employed to find three principal components of the point set $G_i^0$, and each point cloud

$G_{i_j}^0$ was projected to the three principal components to generate a new point cloud, whose first two coordinate components, denoted as $G_{i_j}^1$, could be used to describe the $j$th shape of the $i$th stalk in the plane, and a new point set $G_i^1 = \left\{ G_{i_0}^2, G_{i_1}^2, \cdots, G_{i_j}^2, \cdots \right\}$ could be obtained. Next, rotation transformation was performed on $G_i^1$ to make the fitted straight line of $G_{i_0}^1$ in vertical pose, and the origin of deflection curve was determined by the fitted straight line and the length of the stalk. Subsequently, the point set obtained after rotation was translated to let the origin of deflection curve coincide with the origin of the coordinate system. Finally, reflection transformation might need to be performed to make the rightwards deflection directions of all subsets, and the final obtained point set was denoted as $G_i^2 = \left\{ G_{i_0}^2, G_{i_1}^2, \cdots, G_{i_j}^2, \cdots \right\}$, which could be used to get deflection curves of the $i$th tea stalk.

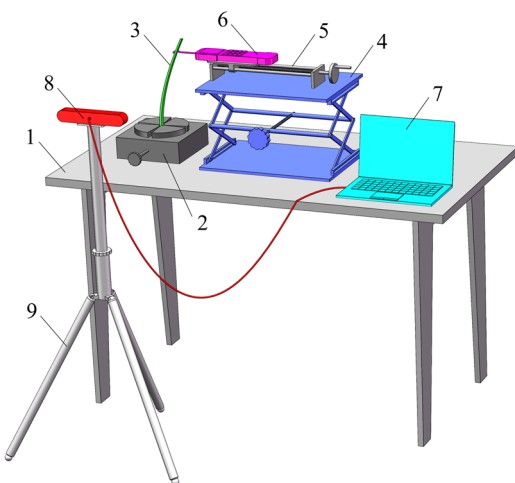

**Figure 1.** Experimental system. (1) horizontal testbed; (2) clamp; (3) stalk; (4) lift platform; (5) horizontal hand-operated stand; (6) digital force meter; (7) personal computer; (8) ZED 2 camera; (9) tripod.

2.2.2. Deflection Curves Determined by Non-Uniform Rational B-Spline Curve Fitting

Any $G_{i_j}^2$ ($j \geq 1$) is fitted into a NURBS curve, which is regarded as observed deflection curve in this study. A $p$th-degree NURBS curve is defined by [27]:

$$
\begin{aligned}
C(u) &= \frac{\sum\limits_{i=0}^{n} N_{i,p}(u)\omega_i P_i}{\sum\limits_{i=0}^{n} N_{i,p}(u)\omega_i}, \qquad 0 \leq u \leq 1 \\
N_{i,0}(u) &= \begin{cases} 1, & | \quad u_i < u < u_{i+1} \\ 0, & |\text{otherwise} \end{cases} \\
N_{i,p}(u) &= \frac{u-u_i}{u_{i+p}-u_i} N_{i,p-1}(u) + \frac{u_{i+p+1}-u}{u_{i+p+1}-u_{i+1}} N_{i+1,p-1}(u) \\
U &= \left[ \underbrace{a, \cdots, a}_{p+1}, u_{p+1}, \cdots, u_n, \underbrace{b, \cdots, b}_{p+1} \right],
\end{aligned}
\tag{1}
$$

where $C(u) = (Cx(u), Cy(u))$, which is a vector-valued function of parameter $u$. $Cx(u)$ and $Cy(u)$ can be used to compute $x$-coordinate and $y$-coordinate of any point on the curve. $\omega_i$ is the weight of the control point $P_i$, and $U$ is the knot vector.

In this study, 2nd-degree NURBS curve with three control points, $\{P_0, P_1, P_2\}$, is employed to fit deflection curves of tea stalks. The knot vector is set as $U = [0, 0, 0, 1, 1, 1]$, and the NURBS curve is also known as a rational Bezier curve in this case. Three control points are set as $P_0 = [0, 0]$, $P_1 = [0, \lambda \cdot y_A]$, and $P_2 = [x_A, y_A]$, respectively, where $y_A$ is the appearance height of the deflection curve, as $\lambda$ and $x_A$ are unknowns. The weights

are set as $w_0 = 1$, $w_2 = 1$, and $w_1$ is unknown. Here, three unknowns are written in a vector form as $X_1 = [\lambda, x_A, w_1]$, which must be solved for fitted NURBS curves of bending shapes of tea stalks. The three unknowns can be determined by solving a least squares problem, whose objective was to minimize the sum of squared residuals (SSR) between observed points in $G_{i_j}^2$ $(j \geq 1)$ and fitted points on the fitted NURBS curve. Specifically, the objective function of the optimization problem can be expressed as:

$$
\begin{aligned}
\min \quad & g_1(X_1) = \sum_{k=1}^{m} |Cx(X_1, u_k) - x_k|^2 \\
s.t. \quad & 0 < X_1(1) < 1 \\
& 0 < X_1(2) \leq \max_{1 \leq k \leq m} x_k \\
& 0 < X_1(3)
\end{aligned}
\tag{2}
$$

where $x_k$ is the $x$-coordinate of the $k$th point in the $G_{i_j}^2$ $(j \geq 1)$, which consists of $m$ points. $u_k$ is the parameterized value of the $k$th point, and its value is solved by the $y$-coordinate of the point with $Cy(u)$. Furthermore, a hybrid particle swarm and simulated annealing stochastic (PSO-SA) optimization algorithm [28] was applied to find a vector $X_1$ that minimizes $g_1(X_1)$ subject to given constraints.

### 2.3. Stalk Model Based on Virtual Internodes

To analyze the bending behavior of tea stalks under the condition of large deflection, the theory of mechanics of material relevant to the cantilever beam was used. For avoiding the complicated impact of the initial shape of the stalk on the investigation, initial axial line of the stalk is assumed to be straight. As shown in Figure 2, the deflection is the displacement in the $x$-direction of any point on the axis of the stalk, and the deflection curve could be expressed as a function of the $y$-coordinate. Besides, the rotation angle of the deflection curve is defined as the angle between $y$-axis and the tangent to the deflection curve.

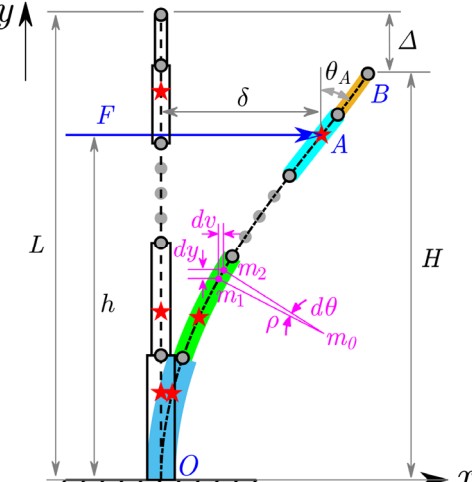

**Figure 2.** Bending diagram of tea stalk in large deflection. $O$, $A$, and $B$ are the fixed point, load point, and free end on the deflection curve of the stalk, respectively. $F$ is the load acting on the height $h$, and $\delta$ and $\theta_A$ are the deflection and rotation angle at the point $A$, respectively. $L$ is the length of the stalk, and $\Delta$ is the vertical displacement at the free end, and $H$ is the appearance height of the stalk after deformation. The dashed line (–) and dash-dotted line (-.) are initial axial line and deflection curve of the stalk, and the circles (o) and pentagrams (★) represent the actual nodes and virtual nodes, respectively.

### 2.3.1. Equations of Deflection and Arc Length

In Figure 2, the points $m_1$ and $m_2$ are located on the deflection curve, and the increments between the two points are $dy$ and $dv$ along the $y$-direction and $x$-direction, respectively. $\theta$ is the rotation angle at $m_1$, and $d\theta$ denotes the increment of rotation angle moved from $m_1$ to $m_2$. Considering the situation that $m_2$ is infinitely close to $m_1$, $m_0$ and $\rho$ are the centre of curvature and the radius of the curvature of the infinitesimal arc, respectively. The arc length $m_1 m_2$ can be calculated as $ds = \rho d\theta = \sqrt{dy^2 + dv^2}$, and the relationship $dv/dy = \tan \theta$ could be used to compute the slope at $m_1$. The exact expression of curvature $\kappa$ at $m_1$ is:

$$\kappa = \frac{1}{\rho} = \frac{d\theta}{ds} = \frac{1}{\sqrt{1 + (dv/dy)^2}} \cdot \frac{d\theta}{dy} = \cos \theta \cdot \frac{d\theta}{dy}. \tag{3}$$

The moment-curvature equation of the bending stalk at $m_1$ can be expressed as a singularity function:

$$\kappa = \frac{M}{EI} = \frac{F[(h-y) + \langle y - h \rangle]}{D} = \begin{cases} \frac{F(h-y)}{D} & , \quad | \, 0 \le y \le h \\ 0 & , \quad | \, y > h \end{cases}, \tag{4}$$

where $M$, $E$, $I$, and $D$ are the bending moment, elastic modulus, moment of inertia, and flexural rigidity, respectively. Based on the above analysis, the deflection curve of the stalk is evidently a piecewise curve if the flexural rigidity varies along the axial line of the stalk. Specifically, $OA$ of the deflection curve is curvilinear, while $AB$ of the deflection curve is an oblique straight line. $OA$ of the deflection curve is the focus in this study. Moreover, flexural rigidity $D$ is selected as the analysis variable, rather than separately investigating the elastic modulus $E$ and moment of inertia $I$.

Combining Equations (3) and (4), and using the method of separation of variables, the following equation can be obtained:

$$\cos \theta \cdot d\theta = \frac{F[(h-y) + \langle y - h \rangle]}{D} \cdot dy. \tag{5}$$

Two mechanical models for plant stalks have been used in previous studies. One of them regarded a plant stalk as a homogeneous prismatic beam with constant flexural rigidity, while the other one regarded a plant stalk as a non-prismatic beam based on actual internodes of the plant stalk.

Here, we developed the mechanical model of tea stalks based on non-prismatic beam with virtual internodes, in which the essential concepts of virtual internode and virtual node were introduced to distinguish from actual internodes and actual nodes. Virtual nodes are defined as points located on an axial line of the stalk and can be different from actual nodes, and virtual internode is defined as the part between two adjacent virtual nodes. With the model, the stalk can be freely divided into multiple virtual internodes by virtual nodes. Coordinate of the $j$th virtual node is denoted as $(v_j, y_j)$ $(j = 1, 2, \cdots, N)$, and the coordinate of $O$ is denoted as $(v_0, y_0)$ for the convenience of following descriptions. Besides, $D_j$ is the flexural rigidity of the $j$th virtual internode, whose $y$-coordinate is between $y_{j-1}$ and $y_j$.

For any point $Q$ on the $j$th virtual internode of the deflection curve, considering the definite integral:

$$\int_{\theta_{j-1}}^{\theta_Q} \cos \theta d\theta = \int_{y_{j-1}}^{y_Q} \frac{F[(h-y) + \langle y - h \rangle]}{D_j} dy \tag{6}$$

where $\theta_Q$ and $y_Q$ are the rotation angle and $y$-coordinate at the point $Q$, respectively. Hypothesize that the rotation angle is continuous along the deflection curve, and $\theta_0$ (the rotation angle at $O$) is $0°$. Thus,

$$\sin\theta_Q = \begin{cases} \frac{F}{D_j}\left(hy_Q - \frac{y_Q^2}{2}\right) & , \quad | \, j = 1 \\ \frac{F}{D_j}\left(hy_Q - \frac{y_Q^2}{2}\right) + F\sum_{i=1}^{j-1}\left(\frac{1}{D_i} - \frac{1}{D_{i+1}}\right)\left(hy_i - \frac{y_i^2}{2}\right) & , \quad | \, j \geq 2 \end{cases} \tag{7}$$

Therefore, the deflection at $Q$ can be calculated as:

$$v_Q = \int_0^{y_Q} \tan\theta \, dy = \int_{y_{j-1}}^{y_Q} \frac{\sin\theta}{\sqrt{1-\sin^2\theta}} dy + \sum_{i=1}^{j-1} \int_{y_{i-1}}^{y_i} \frac{\sin\theta}{\sqrt{1-\sin^2\theta}} dy. \tag{8}$$

Similarly, the arc length $\overparen{OQ}$ can be calculated as:

$$\overparen{OQ} = \int_0^Q \frac{1}{\cos\theta} dy = \int_{y_{j-1}}^{y_Q} \frac{1}{\sqrt{1-\sin^2\theta}} dy + \sum_{i=1}^{j-1} \int_{y_{i-1}}^{y_i} \frac{1}{\sqrt{1-\sin^2\theta}} dy. \tag{9}$$

In combination with composite Simpson's rule, the values of $v_Q$ and $\overparen{OQ}$ could be approximated as Equations (10) and (11), respectively.

$$v_Q \approx \sum_{i=1}^j \frac{\alpha_i}{3n}\left[f_1(y_{i,0}) + 2\sum_{t=1}^{\frac{n}{2}-1} f_1(y_{i,2t}) + 4\sum_{t=1}^{\frac{n}{2}} f_1(y_{i,2t-1}) + f_1(y_{i,n})\right], \tag{10}$$

$$\overparen{OQ} \approx \sum_{i=1}^j \frac{\alpha_i}{3n}\left[f_2(y_{i,0}) + 2\sum_{t=1}^{\frac{n}{2}-1} f_2(y_{i,2t}) + 4\sum_{t=1}^{\frac{n}{2}} f_2(y_{i,2t-1}) + f_2(y_{i,n})\right], \tag{11}$$

where $\alpha_i = y_i - y_{i-1}$ ($i = 1, \cdots, j-1$) while $\alpha_i = y_Q - y_{j-1}$ ($i = j$), and $y_{i,t} = y_{i-1} + \frac{t\alpha_i}{n}$ ($t = 1, \cdots, n$). Besides, $f_1 = \frac{\sin\theta}{\sqrt{1-\sin^2\theta}}$, and $f_2 = \frac{1}{\sqrt{1-\sin^2\theta}}$. Then, $n$ must be an even number, and the value of $n$ in Equations (10) and (11) is set to 8.

### 2.3.2. Calculation Method for Flexural Rigidity

If the deflection curve from $O$ to $A$ of tea stalk is divided into $N$ virtual internodes, the values of flexural rigidities $\{D_j \, | \, j = 1, 2, \cdots, N\}$ of all virtual internodes can be calculated by combining the developed model with the observed deflection curve (fitted 2nd-degree NURBS curve). However, it is almost impossible to solve piecewise flexural rigidities directly by the observed deflection curve and exact deflection equation, i.e., Equation (8). Additionally, how to determine the positions of virtual nodes is an inevitable problem. In this study, virtual nodes are determined by artificially giving the lengths, $L_1$, $L_2, \cdots, L_N$, of $N$ virtual internodes.

As shown in Figure 3a, a method was proposed to solve the flexural rigidity of tea stalk. Firstly, the deflection curve from $O$ to $A$ is fitted into a NURBS curve as the method described in Section 2.2, and arc length $\overparen{OA}$ is calculated simultaneously. Secondly, the NURBS curve is divided into $N$ virtual internodes according to the given lengths of all virtual internodes, meanwhile, the virtual nodes are determined on the NURBS curve. Next, $ys$, a set of $y$-coordinates, is generated between 0 and the load height $h$, and deflections at $ys$ and arc lengths from 0 to $ys$ are calculated by the NURBS curve and the developed model, respectively. Finally, piecewise flexural rigidities can be determined by solving an optimization problem, whose objectives are minimizing not only the SSR between observed deflections and calculated deflections but also the SSR between observed arc lengths and calculated arc lengths. Observed deflections and observed arc lengths are computed at $ys$ by the fitted NURBS curve, while the calculated deflections and calculated arc lengths are computed by the developed model.

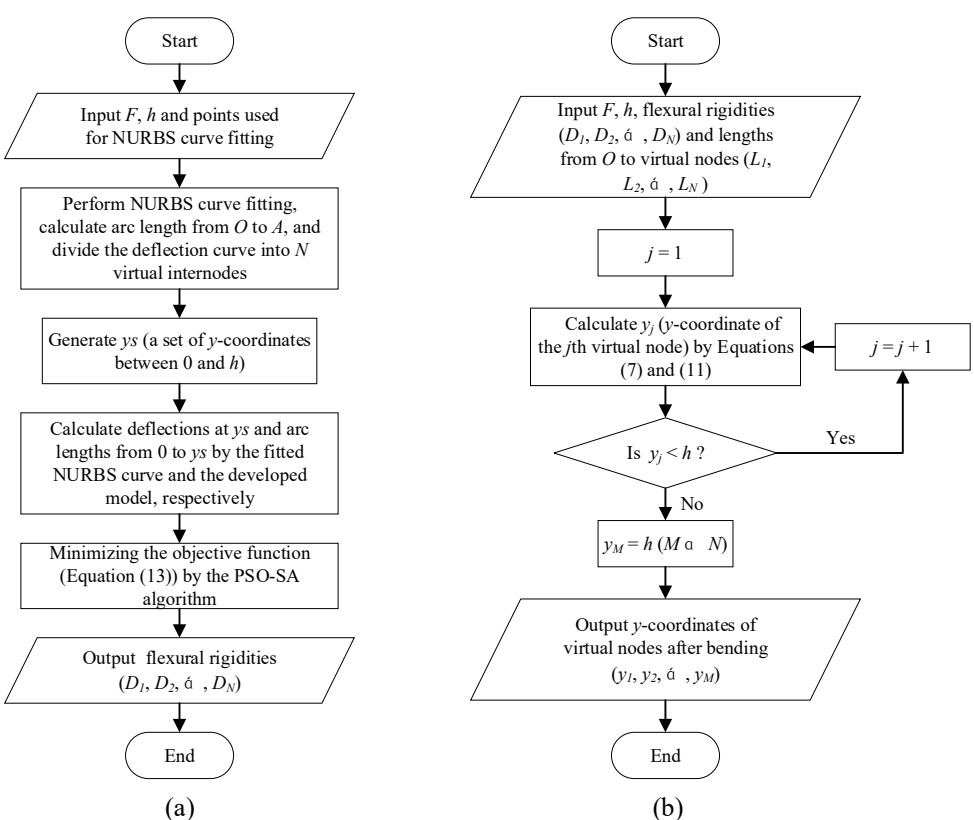

**Figure 3.** (**a**) The flow chart of calculation for piecewise flexural rigidities. (**b**) The flow chart of calculation for the *y*-coordinates of virtual nodes after bending.

For determining the positions of virtual nodes on the NURBS curve according to the given lengths of *N* virtual nodes, the integral related to the first derivatives of NURBS curve was employed. The first derivatives of NURBS curve can be denoted as $C'(u) = (Cx'(u), Cy'(u))$, and the arc length $\overarc{OD}$, from *O* to any point *D* on the NURBS curve, can be calculated as:

$$\overarc{OD} = \int_0^{u_D} \sqrt{(Cx'(u))^2 + (Cy'(u))^2}\,du. \tag{12}$$

Based on the equation, $u_D$ can be solved for a given $\overarc{OD}$, and the coordinate of *D* can be calculated as $[Cx(u_D), Cy(u_D)]$.

To determine *ys*, $T$ $(T > 2)$ equally spaced *y*-coordinates, including *y*-coordinates of two adjacent virtual nodes, are calculated in each virtual internode, and $(T-1)N$ different *y*-coordinates, excluding 0, can be obtained in total.

The objectives of the optimization problem are expressed as an objective function:

$$\min g_2(X_2) = \sum_{k=1}^{(T-1)N} \left| def_k^{obs} - def_k^{cal}(X_2) \right|^2 + \sum_{k=1}^{(T-1)N} \left| arc_k^{obs} - arc_k^{cal}(X_2) \right|^2, \tag{13}$$

where $X_2$ is an unknown vector composed of all flexural rigidities, i.e., $X_2 = [D_1, \cdots, D_N]$. $def_k^{obs}$ and $arc_k^{obs}$ are the observed deflection and observed arc length at the *k*th value in *ys* computed by Equations (1) and (12), respectively. While $def_k^{cal}(X_2)$ and $arc_k^{cal}(X_2)$ are calculated deflection and calculated arc length at the *k*th value in *ys* computed by Equations (10) and (11), respectively. Furthermore, the objective function is solved by the PSO-SA algorithm, and all flexural rigidities could be determined simultaneously.

### 2.3.3. Calculation Method for Deflection Curve

The deflection curve of a tea stalk with certain flexural rigidity at given virtual internodes could be predicted when giving a horizontal load at certain height in the case of elastic deformation. In fact, the deflection of any point on deflection curve can be calculated approximately by Equations (7) and (10) if $y$-coordinates of virtual nodes after bending are determined. Figure 3b shows the flow chart to solve $y$-coordinates of virtual nodes under given load conditions. The key is to solve $y$-coordinates of each virtual node one by one with Equations (7) and (11). It is worth nothing that load point $A$ must be regarded as the last virtual node, which can be any point on deflection curve rather than one of virtual nodes. The value of $y_M$ is the height at load point, and $M$ ($M \leq N$) is the number of virtual internodes contained in the deflection curve for the bending shape.

## 3. Results and Discussion

Lengths of tea stalks selected for the experiment ranges from 170 to 290 mm with 5 to 9 internodes. Following sections presented some results of fitted NURBS curves of tea stalks and illustrated how to solve flexural rigidity of the stalk and predict deflection curve under given load conditions based on the developed model with virtual internodes.

### 3.1. Fitted Non-Uniform Rational B-Spline Curves for Bending Shapes of Tea Stalks

In total, 120 point sets were obtained by the method described in Section 2.2.1, including 6 point sets of each stalk with 1 for the initial shape and 5 for the bending shapes. Figure 4 showed an example of a fitted line for the initial shape and a fitted NURBS curve for the bending shape of a tea stalk, respectively. Concretely, the fitted line for the initial shape is $x = 0$, and the deflection curve for the bending shape was fitted into a 2nd-degree NURBS curve.

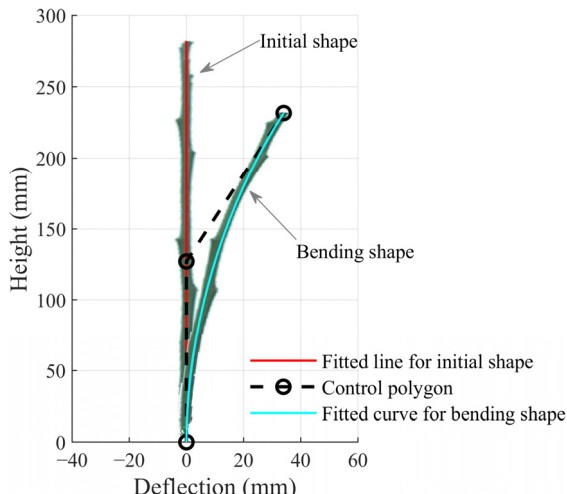

**Figure 4.** An example of curve fitting for tea stalk.

The points of tea stalks for the initial shape ranges from 3027 to 7063 with an average of 4894, and the values of coefficient of determination ($R$-squared) of the fitted NURBS curves for the point sets $\left\{ G_{i_0}^1 \mid i = 1, \cdots, 20 \right\}$ ranges 0.9769 from 0.9937 with an average of 0.9881. Besides, the points of tea stalks for bending shape ranges from 1746 to 5849 with an average of 3523, and $R$-squared values of fitted NURBS curves ranged from 0.9576 to 0.9964 with an average of 0.9797. The frequency distributions of the number of points of stalks for bending shape and $R$-squared values of fitted NURBS curves were shown in Figure 5a and b, respectively.

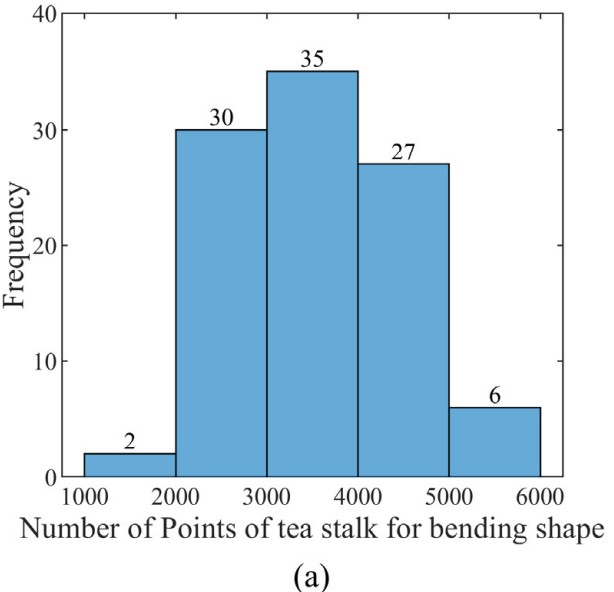

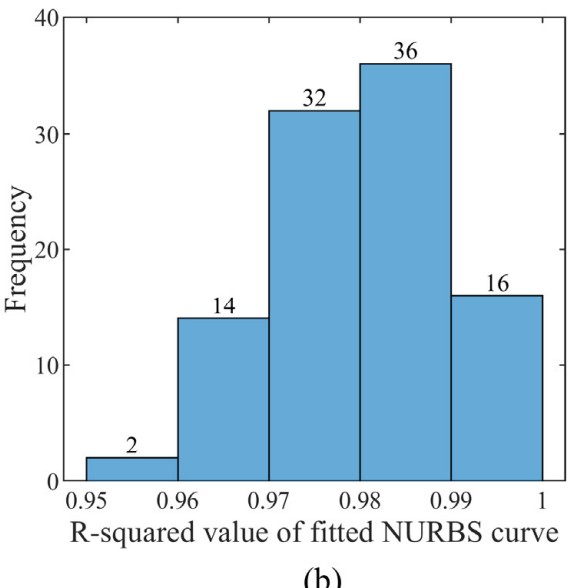

(a)

(b)

**Figure 5.** (**a**) Histogram of the number of points of tea stalks for bending shape; (**b**) Histogram of the *R*-squared value of fitted NURBS curves of tea stalks for bending shape.

Deflection curve reflects the bending characteristics of plant stalks. The existing studies usually provide methods to obtain positions of several discrete points such as the load point and stalk's nodes, hence it limits the investigation on bending properties such as the flexural rigidity of plant stalks. To deeply study the bending characteristics of plant stalks, it is necessary to obtain the deflection curve of plant stalks. In the study, a novel method was proposed to get the deflection curve of tea stalks. The method includes two important steps: (1) obtain the point set $G_{i_j}^2$ ($j \geq 1$) describing the bending shape of tea stalk in the plane, and (2) perform 2nd-degree NURBS curve fitting to obtain the deflection curve of tea stalk. The 2nd-degree NURBS curve used in this study has some advantages: (1) three control points are directly related to geometric shape of deflection curve. Specifically, two control points are two end points of deflection curve, and they also limit the value range of the other control point; (2) the fitted NURBS curve contains information of all points in point set $G_{i_j}^2$ ($j \geq 1$), but only three unknowns ($\lambda$, $x_A$, $w_1$) need to be solved for each bending shape of tea stalk; and (3) all fitted NURBS curves of different bending shapes of the same tea stalk are intersected at a fixed point (origin of the coordinate system), and the angles are 0° between *y*-axis and the tangent lines at fixed point of fitted curves.

### 3.2. Calculation of the Flexural Rigidity of Tea Stalks

Figure 6 shows an example of the calculated average flexural rigidity and piecewise flexural rigidities according to the method presented in the Figure 3a. The arc length of the NURBS curve was 239.36 mm and was divided into four virtual internodes, whose lengths of the first three virtual internodes were 60 mm, while the length of the last virtual internode was less than 60 mm. The average flexural rigidity and piecewise flexural rigidities are displayed in Figure 6, and all digits after decimal point are not displayed in all figures. Using the average flexural rigidity and piecewise flexural rigidities respectively, 30 deflection points were calculated on each virtual internode and a total of 120 deflection points were obtained for each case. The maximum errors between the observed deflections and calculated deflections were −5.21 and 0.26 mm, respectively, and the root-mean-square errors (RMSE) were 1.78 and 0.05 mm, respectively.

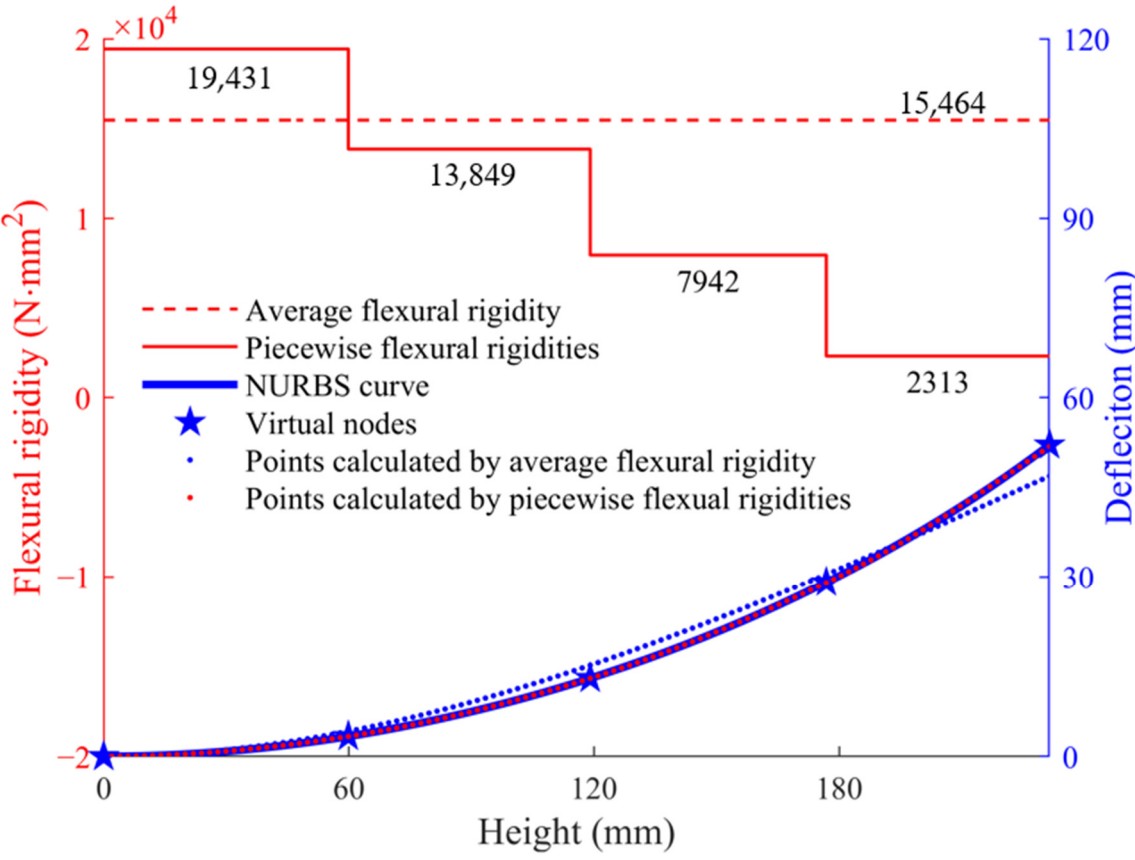

**Figure 6.** An example of average flexural rigidity, piecewise flexural rigidities, and their corresponding deflection curves calculated by the developed model based on virtual internodes.

The impact of length of virtual internode on flexural rigidity and deflection error have been assessed. As illustrated in Figure 7, the values of flexural rigidity and deflection error were calculated under three different situations, namely three lengths (40, 60, and 80 mm) of the virtual internode, in which the tea stalk was divided into 6, 4, and 3 virtual internodes, respectively. Additionally, 30 points were calculated on each virtual internode.

The difference was in the low double digits among the calculated values of the average flexural rigidity in the three situations, and its mean value was displayed in Figure 7a. Apparently, a larger error occurred on the deflections calculated by the average flexural rigidity than piecewise flexural rigidities, especially on the load point. Significantly, the deflections calculated by the average flexural rigidity were first larger and then smaller than the observed deflections as the length increased, as shown in Figure 7b. More specifically, the deflection error increased from zero at first and then decreased to negative. Both phenomena demonstrated that the actual flexural rigidity decreased from the bottom to the top of the stalk. In all three situations, a downward trend of the flexural rigidity always appeared as the increase of length. The maximum deflection errors calculated by these piecewise flexural rigidities were smaller than 0.5 mm, and the shorter the length of virtual internode was, the smaller the RMSE value of the deflection error would be. Additionally, a relative larger deflection error appeared near the top end of the stalk. Similar results also occurred on other experimental tea stalks.

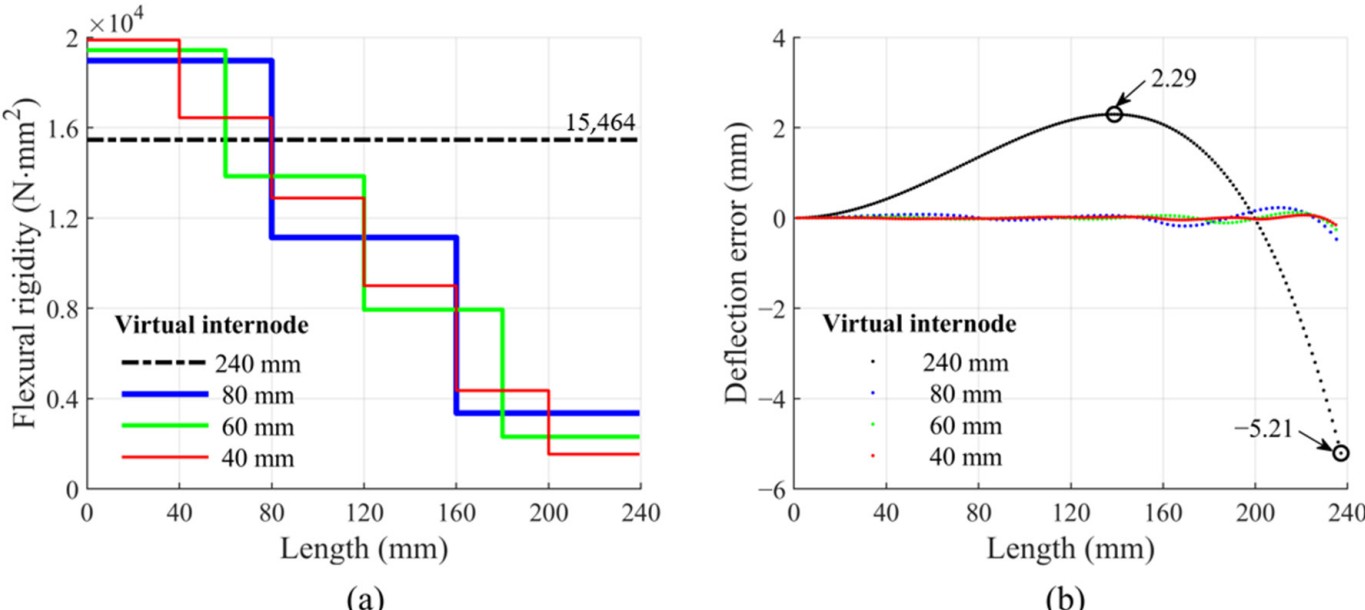

**Figure 7.** (**a**) Impact of length of virtual internode on flexural rigidity. (**b**) Impact of length of virtual internode on deflection error. The positive (negative) value represents that the calculated deflection is larger (smaller) than the observed deflection. The abscissa of two coordinate frames represents the length along the axial line of tea stalk.

To calculate the flexural rigidity of the plant stalks, they are usually regarded as either a prismatic beam with constant flexural rigidity or a non-prismatic beam, in which each internode of the plant stalks is regarded as a prismatic beam with constant flexural rigidity. As shown in Figure 8a and 8b, the whole stalk or the stalk's segment was bended through a three-point bending test or cantilever test, and the average flexural rigidity was calculated according to the given load and deflection at load point by the approximate equations of the deflection curve of the simply supported beam or cantilever beam. The increase of deflection at the load point and length of test specimen would result in greater error in the calculated average flexural rigidity. To obtain the flexural rigidity of the crop stalks, Hirai et al. [20] proposed an interesting calculation method, in which the flexural rigidities of all actual internodes were computed one-by-one based on the coordinates of each actual node, as shown in Figure 8c. However, the flexural rigidity of an internode is very sensitive to the accuracy of the corresponding node's coordinates, which is defined as the center coordinates of a black marker bonded to the node. Considering the measurement error, the obtained nodes may not be precisely located on the actual deflection curve. Additionally, a large error is likely to occur if only a node is considered when calculating the flexural rigidity of an internode, and the error would directly influence the subsequent calculations.

Generally, the diameter and maturity of tea stalk decreases from the lower end to the upper end, and it is harder to bend the lower region than the upper region. Intuitively, the flexural rigidity of the tea stalk decreases upward, and the trend may appear even at the same internode. Moreover, a large deflection usually occurs during the interaction between tea stalks and tea picking machines. Considering the shortcomings of traditional methods and the bending characteristics of tea stalks, the mechanical model based on a non-prismatic beam with virtual internodes was developed, by which the tea stalk can be freely divided into multiple virtual internodes by the given virtual nodes, which could be different from the actual nodes. The application of virtual nodes eliminates the limitations of actual nodes and therefore largely improves the flexibility of the model. Theoretically, the average flexural rigidity of any part, not only the actual internodes, of the plant stalk could be obtained. Flexural rigidities of all virtual internodes are determined simultaneously by solving an optimization problem, in which more points, not only virtual nodes, but

also multiple equally spaced points in each virtual internode, as shown in Figure 8d, are considered.

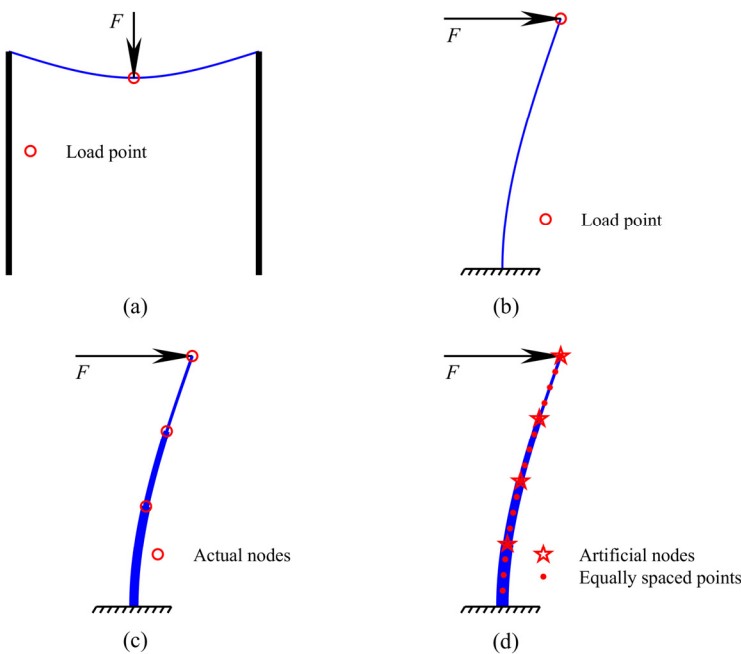

**Figure 8.** Different models used for calculating the flexural rigidity of plant stalks. (**a**) Simply supported beam with a load point. (**b**) Cantilever beam with a load point. (**c**) Non-prismatic cantilever beam with actual nodes. (**d**) Non-prismatic cantilever beam with virtual nodes and equally spaced points in virtual internodes.

*3.3. Deflection Prediction of Tea Stalks*

According to the flow chart described in Figure 3b, the deflection curve of tea stalk can be calculated approximately. Figure 8a showed an example of the observed deflection curves and predicted deflection points of two bending shapes under given horizontal loads, using the same tea stalk as shown in Figure 6. Specifically, 51 equally spaced heights from fixed point to load point were generated to calculate their deflections by the developed model of average flexural rigidity and that of piecewise flexural rigidities (the length of virtual internode was 60 mm), respectively. As shown in Figure 9b, for shape 1, the maximum errors in deflection were 2.03 and 1.42 mm respectively, and the corresponding RMSE values of the deflection error were 1.27 and 0.64 mm, respectively; for shape 2, the maximum errors in deflection were 3.27 and −0.87 mm, respectively, and the corresponding RMSE values of deflection error were 1.98 and 0.51 mm, respectively. Generally, the deflection calculated by the developed model of piecewise flexural rigidities is much closer to the observed deflection than that calculated by the developed model of flexural rigidities.

It was noticed that the observed deflections were smaller than the calculated deflections for Shape 2 using the average flexural rigidity, which indicated that the actual average flexural rigidity of shape 2 was larger than the used average flexural rigidity. This phenomenon may be explained from two hands. On the one hand, the flexural rigidity decreases from the bottom to the top of the stalk. On the other hand, the arc length of shape 2 was shorter than the arc length corresponding to the used average flexural rigidity. According to the NURBS form expressions of the observed curves, the former was 193.37 mm, while the latter was 239.36 mm.

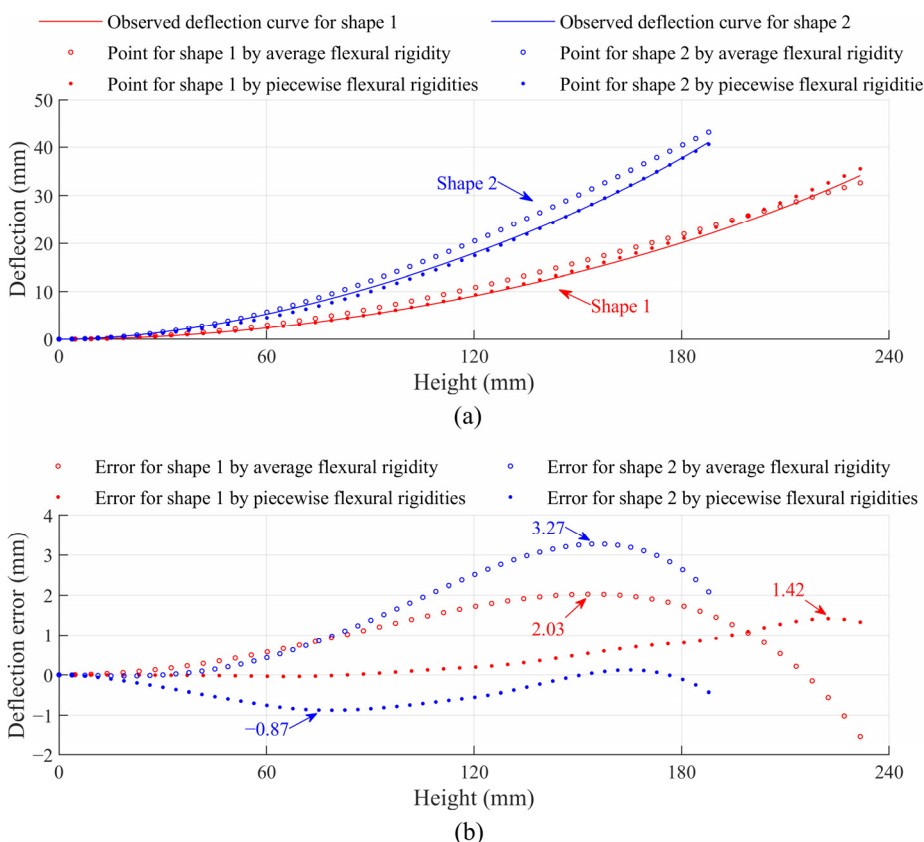

**Figure 9.** (**a**) Observed deflection curves and calculated points for two bending shapes of a tea stalk. (**b**) Errors between observed deflections and calculated deflections for two bending shapes of a tea stalk.

### 3.4. Analysis of the Flexural Rigidity of Tea Stalks

To analyze the influence of variety on the flexural rigidity, we calculate the average flexural rigidity and piecewise flexural rigidities of stalks of *Zhongcha* 108 and *Maolv*. Table 1 presented part of the results, and the first four samples are stalks of *Zhongcha* 108 while the last four samples are stalks of *Maolv*. The flexural rigidity was calculated by the bending shape, whose load point was located on the third actual internode of the stalk. Each stalk was evenly divided into 4 virtual internodes, and 30 equally spaced *y*-coordinates were generated to calculate the deflections in every virtual internode when solving the flexural rigidity. $RMSE_1$ and $RMSE_2$ are the RMSE values of the deflection error at the generated *y*-coordinates for the mean value and piecewise flexural rigidities. The results showed that the average flexural rigidity changes largely for all the samples of the stalk. The calculated piecewise flexural rigidities illustrates that the flexural rigidity decreased from the bottom to the top region of the stalk, and they can calculate the deflection curve more accurately than the mean value. Qualitatively, there are no significant differences between the two varieties of tea plants.

**Table 1.** Flexural rigidity of tea stalks and RMSE values of deflection error.

| Sample Number | Flexural Rigidity/N·mm$^2$ | | | | | $RMSE_1$ /mm | $RMSE_2$ /mm |
|---|---|---|---|---|---|---|---|
| | Mean | Internode 1 | Internode 2 | Internode 3 | Internode 4 | | |
| 1 | 11,494.95 | 16,564.15 | 7816.08 | 6196.97 | 3470.04 | 1.3987 | 0.0398 |
| 2 | 4180.77 | 5516.34 | 3952.98 | 2065.34 | 220.26 | 3.7169 | 0.4319 |
| 3 | 1396.17 | 1447.84 | 1447.84 | 1138.74 | 128.74 | 0.7410 | 0.4530 |
| 4 | 15,463.91 | 19,412.02 | 13,895.68 | 7903.81 | 2321.98 | 1.7807 | 0.0539 |
| 5 | 8259.88 | 9503.64 | 7907.50 | 4442.77 | 1322.55 | 0.8356 | 0.0213 |
| 6 | 9421.01 | 11,024.98 | 9197.43 | 4562.07 | 1090.37 | 1.0586 | 0.0313 |
| 7 | 14,272.19 | 15,738.78 | 15,389.93 | 8484.48 | 2172.84 | 0.8611 | 0.0450 |
| 8 | 2507.05 | 3167.34 | 2100.04 | 1341.10 | 504.56 | 2.1544 | 0.0512 |

## 4. Conclusions

The study focused on the bending behavior of tea stalk under the condition of large deflection. A method was proposed for determining the deflection curve of tea stalk based on the binocular vision technique and NURBS curve fitting technique. The results show that fitted 2nd-degree NURBS curves could accurately represent the deflection curves of tea stalks. According to the difference in resistance to bending among the different parts of tea stalk, the stalk model based on non-prismatic beam with virtual internodes was developed. By introducing the concepts of virtual internode and virtual node, tea stalk could be freely divided into multiple virtual internodes without the limitations of the actual internodes. With the NURBS curve and the model, the flexural rigidity of tea stalks and the deflection curve under given load can be calculated. The results illustrate that the stalks of two varieties of tea plants (*Zhongcha* 108 and *Maolv*) show similar bending characteristics. The flexural rigidity decreased from the bottom to the top of the stalk, and the flexural rigidity shows large differences among the experimental tea stalks.

The calculation method for flexural rigidity can be used to calculate the flexural rigidity of other plant stalks. The research work can be employed to investigate the relationship between quality indicators (length, breakage ratio of leaves, etc.) of harvested tea shoots and operational parameters such as the cutter's structure, cutting height, forward speed, and cutting frequency of tea picking machines. Therefore, it can be further applied to the optimization design of the cutter and adaptive adjustment techniques for the machine.

Further study should be carried out to obtain the deflection curve of tea stalk with leaves, as well as the real time curve fitting technique. Additionally, it is necessary to study other appropriate methods to determine the positions of virtual nodes of tea stalk. More efficient algorithms should be developed to solve the optimization problems for the NURBS curve fitting and the calculation of flexural rigidity of tea stalk, respectively. Moreover, the developed model could be improved by: (1) considering the curvature of initial shape for curving stalks; (2) considering the impact of the weight of ear or fruits such as rice, wheat, and sunflower stalks.

**Author Contributions:** Conceptualization and methodology, W.W. and Y.H.; software, W.W.; data curation and writing—original draft preparation, W.W.; writing—review and editing, W.W. and Z.J. All authors have read and agreed to the published version of the manuscript.

**Funding:** This research was funded by the Priority Academic Program Development of Jiangsu Higher Education Institutions (PAPD-2018-87).

**Institutional Review Board Statement:** Not applicable.

**Informed Consent Statement:** Not applicable.

**Conflicts of Interest:** The authors declare no conflict of interest.

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
