# Peer review of "Investigation on the Bending Behavior of Tea Stalks Based on Non-Prismatic Beam with Virtual Internodes"

_agriculture, doi:10.3390/agriculture12030370_

Round 1

Reviewer 1 Report

Abstract: at the end of the abstract rather than stating that this model can be applied to other crops I would appreciate to give some hints regarding the practical applications of this study in the optic of actual harvesting of tea stalks.

Keywords: I suggest to avoid keywords already used in the title

Line 26: in this first sentence I think it is needed to give a geographic framework to this assertion: where in the world labor shortage and low mechanization are a problem for tea harvesting?

At the beginning of the introduction it is necessary to report in few lines the current technology to harvest tea, this information is given but in a too direct and confusing way, it is instead a very important part at the base of the study.

Before starting the literature review (line 37) authors should explain why it is important to study bending behavior of plant stalks.

Lines 61-62: small in a scientific context means rather nothing, please provide some thresholds referring to related literature.

At the end of introduction I suggest to report in a clear way what are the practical implications and possible benefits of the study. Ok, it is important to develop a suitable model for the bending behavior of tea stalks but what can be practically done with these results and by who?

Line 111: why two varieties? Or better why authors referred to two different varieties at the beginning of M&M methods and then carried out the study without considering this possible source of variability within their study? If authors add a factor (in this case variety) to the experimental design they have to account for such potential source of variability because it is not possible to exclude that different varieties can have different behavior. Therefore there are different options for authors: 1) repeating the study separately for the two varieties and assess the possible differences (this will give even higher strength to their experimental design and to the validity of their findings), 2) doing preliminary test on the bending behavior of the two varieties and statistically checking that there are no significant differences, in this case the test can be carried out excluding that “variety” is a source of variability for the experimental design, 3) finding and citing literature references which stated that the bending behavior of tea stalks is not influence by tea cultivar, in this case authors can do the assumption that this parameter has no influence for their study without any problem. Without exploring the influence of the parameter “variety” on the results there will be always some doubts about the possible influence of this parameter on the obtained results.

Discussion section has to be completely revised according my thinking: firstly it is rather a summary of authors’ results than a real discussion, indeed only one reference to current literature is given. The discussion section of a scientific manuscript should highlight the results of the study in comparison to the current state of the art in a specific topic not just commenting, moreover in a not deep way, the obtained data. If there are no many references to cite, and sometimes it can happen, authors should however address this issue within the text, explaining why it is like this and therefore why the discussion is much more focused on the obtained results without comparison to current literature. In this case a single results and discussion section is often more suitable.

Moreover it is still not clear what are the practical implications of the study. Considering that the developed model is well functioning how can someone (machine constructors? Researchers? Engineers?) use the model from a practical point of view? If authors developed a model and then this model remains in the hard disk of their computer it is not useful. They should clearly describe the practical (can be future but practical) possible applications of this model.

Reviewer 2 Report

Investigation on Bending Behavior of Tea Stalks Based on Non-prismatic Beam with Virtual Internodes is good work which is about development of deflection curve based on the technqiues of Binocilar vision and non-uniform rational B-spline (NURBS) curve fitting. My query is how this technqiue was evaluted with real time data set. Another query is author are asking to improve the model but the question what aspects future work should be consucted to improve the model.

Round 2

Reviewer 1 Report

Authors deeply revised the manuscript, congratulations.